# The Impact of the Lockdown Caused by the COVID-19 Pandemic on the Fine Particulate Matter (PM_2.5_) Air Pollution: The Greek Paradigm

**DOI:** 10.3390/ijerph18136748

**Published:** 2021-06-23

**Authors:** Ourania S. Kotsiou, Georgios K. D. Saharidis, Georgios Kalantzis, Evangelos C. Fradelos, Konstantinos I. Gourgoulianis

**Affiliations:** 1Respiratory Medicine Department, Faculty of Medicine, University of Thessaly, BIOPOLIS, 41110 Larissa, Greece; kgourg@uth.gr; 2Department of Nursing, Faculty of Nursing, University of Thessaly, GAIOPOLIS, 41110 Larissa, Greece; efradelos@uth.gr; 3Department of Mechanical Engineering, University of Thessaly, Leoforos Athinon, Pedion Areos, 38334 Volos, Greece; saharidis@gmail.com (G.K.D.S.); george.kalantzis4@gmail.com (G.K.)

**Keywords:** air pollution, coronavirus disease 2019, Greece, GreenYourAir, fine particulate matter

## Abstract

Introduction: Responding to the coronavirus pandemic, Greece implemented the largest quarantine in its history. No data exist regarding its impact on PM_2.5_ pollution. We aimed to assess PM_2.5_ levels before, during, and after lockdown (7 March 2020–16 May 2020) in Volos, one of Greece’s most polluted industrialized cities, and compare PM_2.5_ levels with those obtained during the same period last year. Meteorological conditions were examined as confounders. Methods: The study period was discriminated into three phases (pre-lockdown: 7 March–9 March, lockdown: 10 March–4 May, and post-lockdown period: 5 May–16 May). A wireless sensors network was used to collect PM_2.5_, temperature, relative humidity, rainfall, and wind speed data every 2 s. Results: The lockdown resulted in a significant drop of PM_2.5_ by 37.4% in 2020, compared to 2019 levels. The mean daily concentrations of PM_2.5_ exceeded the WHO’s guideline value for 24-h mean levels of PM_2.5_ 35% of the study period. During the strictest lockdown (23 March to 4 May), the mean daily PM_2.5_ levels exceeded the standard 41% of the time. The transition from the pre-lockdown period into lockdown or post-lockdown periods was associated with lower PM_2.5_ concentrations. Conclusions: A reduction in the mean daily PM_2.5_ concentration was found compared to 2019. Lockdown was not enough to avoid severe exceedances of air pollution in Volos.

## 1. Introduction

The Mediterranean city of Volos is located in Central Greece and situated midway on the Greek mainland. It is the sixth-largest city with the third-largest industrial area. The Volos port is the third-largest port in Greece. Volos is an excellent paradigm of a medium-size city where population shifts and high industrialization in the last decades have resulted in the degradation of the air quality [1]. It is considered among the most polluted cities in Greece, primarily due to domestic heating, traffic, container terminal operations, mineral facilities, and installations for cement and lime production [2].

Air pollutants include gaseous pollutants and particle matters (PM). The pathogenicity of PM is determined by their size, origin, composition, solubility, and ability to produce reactive oxygen species. It has been reported that smog is generally caused by high concentrations of aerosols or fine particles sized less than or equal to 2.5 micrometers, referred to as inhalable fine particulate matter (PM_2.5_) [3,4,5,6]. The toxic effects of PMs are mainly attributed to PM_2.5_ [3]. PM_2.5_ pollution is mainly related to anthropogenic emissions from industries, traffic transportation, power plants, and biomass burning [4]. Vehicular traffic constitutes the most important source of particulate pollution in the area under study [1]. More specifically, tourist traffic passing through the city towards the local attractions and the seaport for passengers and commercial use are two factors that aggravate traffic. Among many other sectors, transport is the most hard-hit sector due to lockdown. Road and air transport came to a halt as people were not allowed or hesitate to travel.

The increased industrialization in the city of Volos is strongly related to PM_2.5_ pollution. Cement, steel, and mineral mining industries are making a source of atmospheric pollution in Volos [1,2]. Additionally, there are two relatively small industrial areas to the west and a big cement industry to the east of the study area. Other sources of PM pollutants are the burning of fossil fuels in vehicles and power plants [1,2].

Factors affecting PM_2.5_ mass concentration apart from domestic pollutant emission and external sources [7,8] include the meteorological parameters (relative humidity, wind speed, temperature). These parameters affect pollution concentration, as well as the removal, transportation, and dispersion of airborne particles [9,10,11,12,13]. The climate of Volos is of Mediterranean type with mild, wet winters and hot, dry summers. The average daily temperature for the spring is 15 °C (March, April, May) [1]. The average daily relative humidity varies between 58% in July and 74% in November [14,15,16]. During 7 March–23 May is a period free from significant activity of residential heating equipment in the city. Also, this period is characterized by the absence of abrupt weather changes and the absence of precipitation which would clean the air, so the local wind system and micro-climate are clearly observable [3].

Saharan dust advection days have been shown to contribute to particulate matter exceeding the daily WHO-recommended limits in the city [17,18,19,20,21,22]. The cyclones are generated by the thermal contrast between cold Atlantic air and warm continental air that cross North Africa during spring and summer [17,18,19,20]. Furthermore, the presence of circumferential, mountainous terrain creates local air turbulences, making air exchange problematic and air polluted with PM_2.5_ may circulate over the region—a condition known as the long-range transport of air pollution. Sea salt emissions (whose growth rate depends on relative humidity) are also a source of PMs and contribute up to 80% of particle levels in the air in the coastal area of Volos [5]. Sea salt contributed by 12.4%, as expected for the maritime location of Volos [6].

The PM_2.5_ air pollution has been associated with an increased risk of acute or chronic respiratory disease and susceptibility to exacerbations given that prolonged exposure to air pollution leads to a chronic inflammatory stimulus, even in young and healthy subjects [23,24,25,26,27,28,29,30,31]. Particle air pollution has been associated with more medical visits and excess hospitalizations [29,30,31]. Furthermore, people living in areas with high levels of particular air pollutants are vulnerable to developing respiratory infections [31]. Namely, recent evidence supports that the high pollution level of Northern Italy should be considered an additional co-factor of the high level of COVID-19 death rates recorded in that area [32,33]. Emerging data supporting air pollution exposure is linked to COVID-19 severity, higher morbidity, and mortality [32,33,34]. Utilizing health information in air pollution health research will result in the achievement of environmental health protection goals.

Globally, interventions to contain the coronavirus disease 2019 (COVID-19) outbreak led to improvements in air quality [35]. The coronavirus pandemic and lockdown slowed business activities, restricted traffic and transportation, and revealed a huge drop in air pollution in affected countries [32]. Responding to the ongoing novel coronavirus outbreak, Greece implemented the largest quarantine in the country’s history during the first pandemic wave. A three-phase approach was adopted. On the 10th of March, the operation of educational institutions of all levels was suspended nationwide, and then, on the 13th of March, all commercial stores and entertainment centers were closed down. The Greek authorities announced stringent traffic, transport, and industry restrictions, starting from 6 AM on the 23rd of March. Starting from the 4th of May, after a 42-day lockdown, Greece gradually lifted restrictions on movement and restarted business activity. The measures put in place in Greece were among the strictest in Europe [32,35]. As Greece went into lockdown, the industrial activities shut down in Volos.

No data exist regarding the impact of the lockdown caused by the COVID-19 pandemic on PM_2.5_ air quality in Greece. The aims of this study were to assess the PM_2.5_ concentrations obtained by a wireless sensors network located at twelve different measurement points in Volos, Greece, before (7 March–9 March), during (10 March–4 May), and after the implementation of lockdown (5 May–16 May), examine to what extent government restrictions affected PM_2.5_ concentrations, and compare PM_2.5_ levels with those obtained from the identical locations during the same period in 2019. Meteorological conditions were also examined as confounders.

## 2. Materials and Methods

### 2.1. Network Implementation and Data Collection

The daily 24-h PM_2.5_ air pollution data were collected from twelve fully automated air quality monitoring stations located in the center and the greater area of Volos for the period from 7 March 2020 until 16 May 2020. The selected study period is characterized by the absence of abrupt weather changes and the absence of precipitation which would clean the air, so, the local wind system and micro-climate are clearly observable. Furthermore, this is a period free from significant activity of residential heating equipment in the city [3]. The values of the twelve monitoring devices have been used to calculate the daily mean PM_2.5_ concentration in Volos. The fully automated air quality monitoring network was established by the GreenYourAir research team. The 24-h temperature, relative humidity, rainfall, and wind speed values were also recorded every 2 s along with each PM_2.5_ measurement by the network, for the same period. The higher safe limits for particulates in the air defined as a daily average of 25 µg/m^3^ for PM_2.5_, according to the World Health Organization (WHO) air quality guidelines [36]. More specifically, the GreenYourAir monitoring network consisted of twelve measuring devices (GreenYourAir device 1178/PM_2.5_) (Figure 1) designed and developed by the GreenYourAir research team. The network was developed based on the outcomes of previous EU projects and tests implemented by the team. GreenYourAir projects focus on monitoring the air quality at the city of Volos and especially PM_2.5_, on identifying correlations between medical incidents and levels of PM_2.5_, on quantifying the origins of air pollution in the monitored area, and on suggesting to public authorities and private entities solutions to improve air quality by decreasing the level of PM_2.5_ [37]. The network works 24 h a day and seven days a week since 1 March 2019. Hence, the 2019 data were collected from the same network (GreenYouAir) and devices (GreenYourAir/1178 device) were placed at the same locations. To make day-to-day comparisons between 2019 and 2020 air quality data, we used the means calculated by averaging daily PM levels from all the twelve locations over the sampling duration.

For the data collection, the light-scattering method was utilized. The amount of light scattered by the particles was detected by a photodiode, which translated the signal into electrical pulses. Then, the microprocessor analyzed these signals and calculated the mass concentration based on the amplitude of pulses. The light-scattering method has been widely used in research projects and the development of smart cities. The main parts of the device are: a sensor that provided data for the concertation of PM_2.5_, temperature, and relative humidity, a zero-one integer (I/O) expansion shield, and an Arduino YUN rev. 2. The programming language of the device was C++. The devices were collecting data per second and they were working 24-h per day. In addition, a 3D printed box was designed to install the devices and create the network [20].

The GreenYourAir research team developed a mathematical formula and an optimization model to determine the optimal locations of the twelve sensors and create the sensor network. A mathematical formula was used to divide the city into sub-areas with specific characteristics and determine the number of sensors in each area. An environmental nuisance of 50 squares was calculated based on the source of emissions. In the end, 12 locations with different environmental nuisance were selected in order to represent well the region under study. The data that were introduced as inputs in the mathematical formula were: the five main functional zones of the city, the traffic zones of the city, the existence of bus, train, and boat stations, the presence of schools, parks, the heat sources, and the geomorphological characteristics of the city. In every smaller area created by the formula, a score was assigned regarding the proper number of sensors that should be placed based on its main characteristics.

More specifically, the city of Volos was divided into five functional zones (commercial and recreational zone, high-density residential zone, medium density residential zone, low-density residential zone, and industrial zone) (Figure 2A). The traffic flow of the city was divided into three types according to the density of traffic (high, medium, and low traffic jams), as shown in Figure 2B.

The city was divided into smaller areas and determined the number of sensors of each area as shown in Figure 3. The 1st industrial zone of Volos is located outside the city to the western suburbs, about 6 km from the center. The Lafarge Cement Volos Plant and the ELINOIL Petroleum company are located outside the city to the eastern suburbs, about 3.5 km from the center of the city. The city’s primary sources of heating are oil, natural gas, and fireplaces. The geographical location and the geomorphological features specific to the city were also analyzed.

The required number of sensors at each sub-area was selected by analyzing the existence of parks, main roads, sports facilities, schools, and universities, the sources of heating, the traffic jam, and the zone within the city. Sensor positioning was the following: one sensor was placed at area A, two sensors at area B, three sensors were placed separately at areas C, E, and F.

The sensor placement locations were strategically placed after an optimization model was developed. The research team formalized the problem by means of a mathematical optimization model guided by the following main parameters: the number of sensors that were placed at each sub-area, the distance between the sensors of the same sub-area, the distance between the sensors of different sub-areas, the specific characteristics of each area and the coverage characteristics among different sub-areas (Figure 4A). The detailed optimization model guided the optimal sensor placement (Figure 4B).

GreenYourAir devices 1178/PM_2.5_ had advantages and disadvantages when compared to the dust samplers. The main advantages of the sensors were the simplicity of real-time measurements, the low cost, and the assessment of board geospatial coverage for the area. The main disadvantage of the sensors, in some cases, could be the accuracy of the measurements. To solve that issue, the GreenYourAir research team developed a calibration methodology.

During the development period, the sensing measurements were tested under laboratory conditions. Their performance was compared to reference instruments. The real-time measurements were compared with the reference instruments to increase accuracy. To this end, real-time air monitoring data is now viewable on the website: http://greenyourair.org/ (accessed on 1 June 2021).

### 2.2. Statistical Analyses

Spearman’s correlation was used for correlation analysis between mean daily concentrations of PM_2.5_ and meteorological variables. Multiple linear analysis and Spearman analysis were used to analyze the correlation between mean daily PM_2.5_ concentrations, meteorological variables, and the three phases of the study. To identify differences between two independent groups, an unpaired *t*-test was used. Parametric data comparing three or more groups were analyzed with one-way ANOVA and Tukey’s multiple comparisons test, while non-parametric data were analyzed with the Kruskal–Wallis test and Dunn’s multiple comparison test. A result was considered statistically significant when the *p*-value was <0.05. Data were analyzed and visualized using SPSS Statistics v.23 (Armonk, NY, USA: IBM Corp.) and Tableau (Tableau Software LLC, Seattle, WA, USA), respectively.

## 3. Results

### 3.1. Comparison between the Ambient PM_2.5_ Levels in 2020 and 2019 (7 March 2020 to 16 May 2020)

Daily PM_2.5_ concentrations used in Figure 5 and throughout this study were obtained by averaging 24-h PM_2.5_ measurements monitored at 12 locations in 2019 and 2020. In Volos over the entire sampling period, the mean concentration of PM_2.5_ declined significantly compared with the same 11-week period in 2019 (34.8 ± 11.9 vs. 21.8 ± 9.2, *p* < 0.001), as presented in Figure 5. There was a 37.4% reduction in mean daily PM_2__.5_ levels during the COVID-19 period in Volos.

The percentage of days with a mean daily concentration of PM_2.5_ above the safe limit was 80% in 2019, i.e., almost the entire study period except for the last week of April and the first days of May. The PM_2.5_ levels were 30% to 100% higher than the 24-h threshold of 25 μg/m^3^ set by the WHO air quality guidelines. In 2020, the mean daily concentrations of PM_2.5_ exceeded the safe limit of 35% of the study period.

In both years, a significant downward trend of PM_2.5_ concentrations after the 26th of April was observed. In both years, PM_2.5_ level reductions were significant during consecutive phases. In 2019, PM_2.5_ levels significantly decreased from phase 1 to phase 3 (43.65 ± 7.72 vs. 19.24 ± 6.68, *p* = 0.001) and from phase 2 to phase 3 (36.6 ± 9.72 vs. 19.24 ± 6.68, *p* < 0.001). Similarly, in 2020, PM_2.5_ levels significantly decreased from phase 1 to phase 3 (29.67 ± 10.71 vs. 12.57 ± 3.87, *p* < 0.001), from phase 2 to phase 3 (21.98 ± 7.07 vs. 12.57 ± 3.87, *p* = 0.001) as well as from phase 1 to phase 2 (29.67 ± 10.71 vs. 21.98 ± 7.07, *p* = 0.015).

During the strictest period amid phase 2 (23 March to 4 May 2020), when stringent traffic, transport, and industry restrictions were implemented, the mean daily concentrations of PM_2.5_ exceeded the safe limits in 41% of the days.

### 3.2. Correlations between Meteorological Variables and PM_2.5_ Air Pollution

We found that PM_2__.5_ concentration was negatively correlated with temperature and positively correlated with humidity. No correlation was found between PM_2__.5_ concentration and rainfall or wind speed (Figure 6).

### 3.3. The Impact of Meteorological Variables and Three-Phase Lockdown Approach on PM_2.5_ Air Pollution

A multiple linear regression analysis was conducted with numerical and categorical variables turned into dummy variables. The mean daily temperature, humidity, rainfall, and wind speed, and phase 2 and phase 3 of the study period were used as the independent variables in the prediction of PM_2.5_ air pollution (Table 1). Phase 2 and phase 3 were compared with phase 1, which was used as a reference group. The predictor variables of phase 2 and phase 3 explained 43.5% of the total variance in this regression model. The transition from phase 1 into phase 2 reduced PM_2.5_ levels by 7.694 μg/m^3^ and from phase 1 into phase 3 by 14.453 μg/m^3^. There was no multicollinearity between the explanatory variables.

## 4. Discussion

In this study, for the first time, we assessed the PM_2.5_ concentration before, during, and after lockdown in one of the most polluted cities in Greece, Volos, compared to the same period in 2019. The relationship between PM_2.5_ mass concentration and meteorological conditions was also determined. Our results showed that PM_2.5_ pollution dropped significantly, by 37.4%, during the COVID-19 three-phase period, compared to 2019. The mean daily concentrations of PM_2.5_ exceeded the safe standards of 35% of the study period in 2020, compared to the PM_2.5_ concentration exceeding the limit values over almost the entire study period (80%) in 2019. However, we found that strict lockdown (23 March to 4 May) was not enough to avoid severe exceedances of air pollution in Volos as the mean daily concentrations of PM_2.5_ exceeded the safe limits 41% of the time. In both years, reduction in the PM_2.5_ levels was significant from phase 1 to phase 2, and from phase 1 to phase 3. In both years, a significant downward trend of PM_2.5_ concentrations after the 26th of April was observed. The transition from the pre-lockdown period (phase 1) to the lockdown period (phase 2) and from the pre-lockdown to the post-lockdown period (phase 3) in 2020, contributed to lower PM_2.5_ concentrations, independently of the existing meteorological conditions.

We found that PM_2.5_ concentrations exceeded the limit values over almost the entire study period (80%) in 2019. Our data accorded with earlier observations supporting that average daily PM_2.5_ concentrations exceeded established standard values in the city [1,38]. Previous reports supported a correlation between the number of days exceeding the daily threshold concentration and the annual hospital admission rates for respiratory diseases [1].

However, a significant reduction of 37.4% in mean daily PM_2__.5_ concentration during the 2020 three-phase COVID-19 period in Volos, compared to 2019 was recorded. According to data from the Department of Mechanical Engineering of the University of Thessaly, 55 GPS devices installed in city buses demonstrated that, on average, traffic was decreased by 50% on the main roads of Volos during the lockdown in the study period [39]. Similarly, in the two largest cities in Greece, Athens and Thessaloniki, there was 21% and 27% less traffic in 2020 than in 2019, mainly attributed to restriction measures, respectively [40].

While lockdowns have caused the decline of air pollution, this did not seem to be enough to avoid PM_2.5_ air pollution events in Volos due to an extensive network of stationary (industry, central heating) and mobile air pollutant sources. Although stringent traffic, transport, and industry restrictions were implemented during the lockdown period, the mean daily concentrations of PM_2.5_ exceeded the safe limits 41% of the recording days during the strictest period of the lockdown (23 March to 4 May). The long-range dust incidences and sea salt emissions during spring could be partly attributed to this observation [5,6].

Moreover, in both years, a significant parallel downward trend of PM_2.5_ concentration was observed over the spring period in Volos. In both years, PM_2.5_ level reductions were significant from phase 1 to phase 2 and from phase 2 to phase 3. This finding is consistent with previous studies that showed a good overall agreement in PM_2.5_ concentration trend lines during spring and summer [41]. Previous reports well-defined mean seasonal variation of air pollutants concentrations after examining large periods in Volos [42,43]. The spring-summer minimum is due to the reduced domestic heating emissions and private car traffic during vacations and the intense flow of the “Etesian” winds [42,43].

In our study, PM_2.5_ concentration was negatively correlated with temperature and positively correlated with humidity, consistent with previous reports of our team [7,8,33]. Supportive evidence shows that PM_2.5_ emissions increased exponentially as temperature decreased, suggesting a negative correlation between temperature and PM_2.5_. A possible explanation for this finding is that when the temperature is higher, the air convection at the lower surface is stronger, which benefits the upward transport of PM_2.5_ [33]. An increase in relative humidity could aggravate PM_2.5_ pollution through physical and chemical processes, affecting the gas-to-particle conversion rate and wet or dry deposition [33]. No correlation was found between PM_2.5_ concentration and wind speed in the present study.

The COVID-19 pandemic constitutes a multifactorial problem requiring multifactorial responses. Our study provides first-time data regarding one of the most polluted industrialized cities in Greece. However, it has some limitations that need to be acknowledged. Firstly, our analysis did not include all Greek cities. Secondly, this study did not examine the effect of other air pollutants such as carbon monoxide, nitrogen dioxide, and sulfur dioxide, which may increase the risk of respiratory tract infections. Moreover, the comparisons of air quality between different city points would be an important line of study designed to be subject for future study. Our study was designed to focus specifically on to what extent government restrictions affected PM_2.5_ concentrations in one of the most polluted cities in Greece, Volos.

## 5. Conclusions

Mediterranean urban agglomerates are characterized by relatively high atmospheric pollution due to anthropogenic (urban, industrial) and natural sources (desert dust, biomass burning) and the prevailing atmospheric conditions that favor photochemical production of secondary pollutants. A spring sampling campaign verified a high concentration of PM_2.5_ in the ambient air of the city of Volos in 2019, with a reduction of 37.4% in mean daily PM_2__.5_ concentration; without avoiding severe exceedances during the 2020 COVID-19 period. Long-term monitoring of atmospheric pollution should be carried out in Volos and Mediterranean cities of similar characteristics. Continued epidemiological and experimental studies are needed to evaluate the role of atmospheric pollution in specific populations and provide more critical information for better preparedness policies in cases of pandemics.

## Figures and Tables

**Figure 1 ijerph-18-06748-f001:**
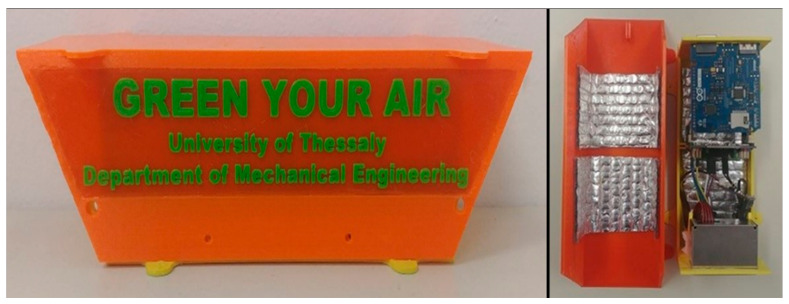
GreenYourAir device 1178/PM_2.5_.

**Figure 2 ijerph-18-06748-f002:**
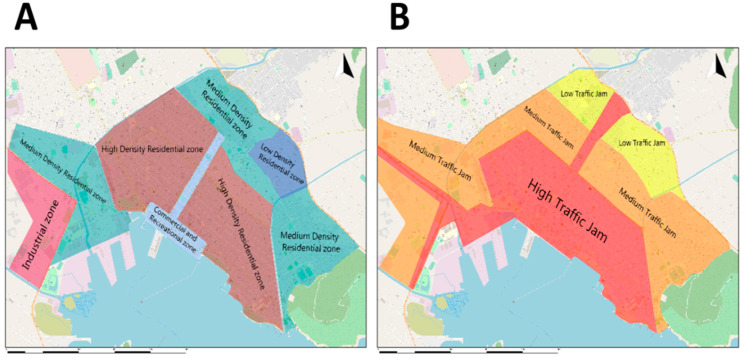
Zones of the city of Volos (**A**) and division into zones according to the density of traffic (**B**).

**Figure 3 ijerph-18-06748-f003:**
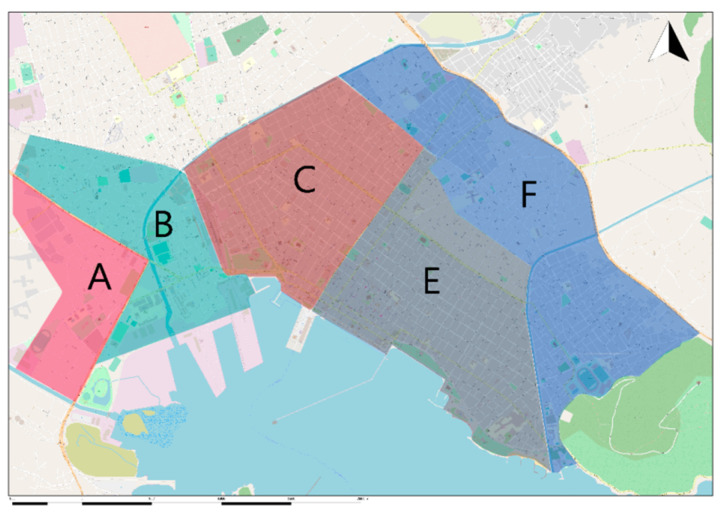
The sub-areas of the city of Volos selected for the placement of the sensors.

**Figure 4 ijerph-18-06748-f004:**
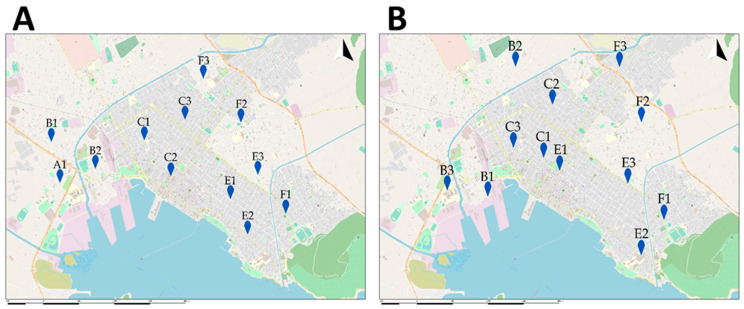
The optimal sensor placement according to the mathematical model (twelve sensors) (**A**) and the actual placement of sensors (twelve sensors) (**B**).

**Figure 5 ijerph-18-06748-f005:**
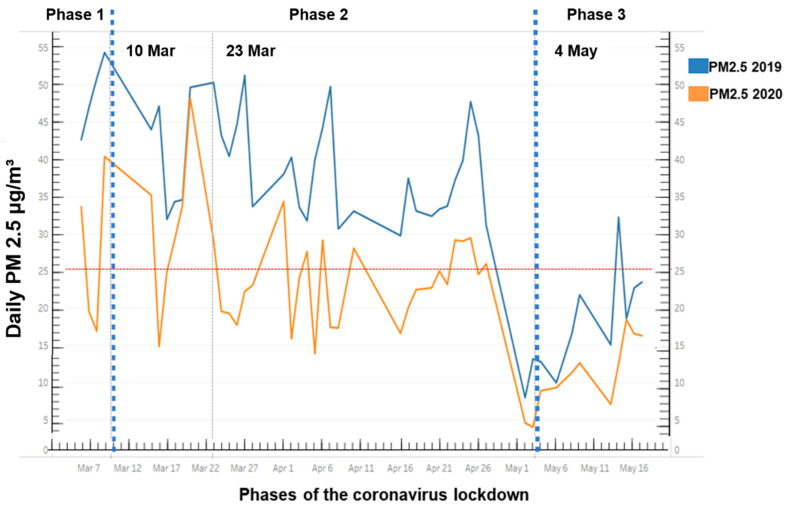
The trend of PM_2.5_ concentration in the city of Volos, Central Greece, from 7 March to 16 May 2020, compared to PM_2.5_ concentrations during the same period in 2019. The horizontal red dashed line identifies the PM_2.5_ concentration at 25 μg/m^3^ set as a safe limit by the World Health Organization air quality guidelines. The dotted dash vertical lines indicate the start of each phase of the three-step approach to combat the COVID-19 wave. Phase 1 (pre-lockdown period): 7 March–9 March, Phase 2 (lockdown period): 10 March–4 May, and Phase 3 (post-lockdown period): 5 May–16 May. The vertical black dashed line represents the initiation of the strictest period amid phase 2 (23 March 2020).

**Figure 6 ijerph-18-06748-f006:**
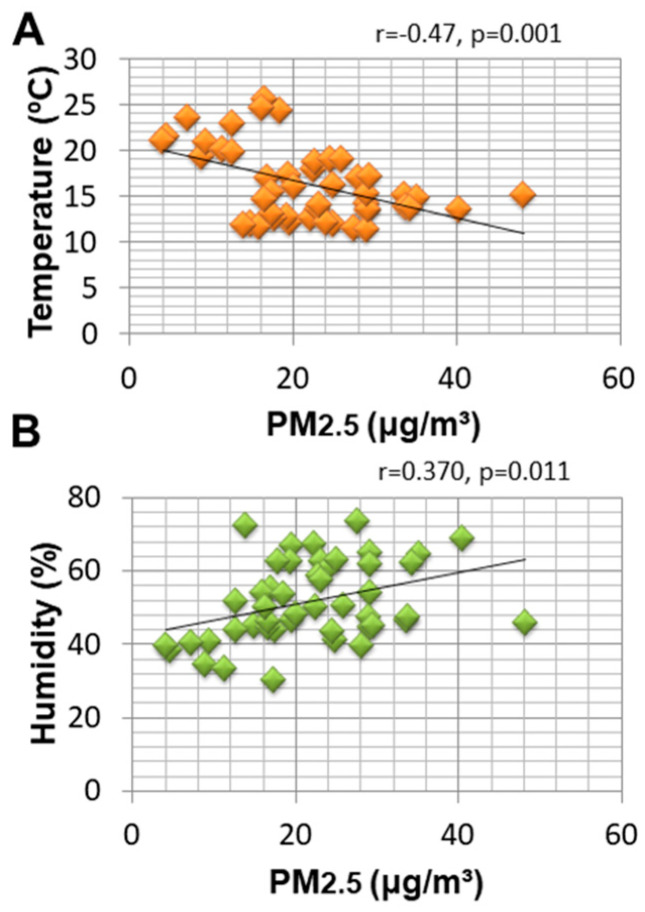
Correlations between meteorological variables and PM_2.5_ air pollution.

**Table 1 ijerph-18-06748-t001:** Multiple linear regression analysis for PM_2.5_ air pollution over the three-phase approach to lifting COVID-19 restrictions in 2020.

Model	Unstandardized Coefficients		Standardized Coefficients	t	Sig.	Collinearity Statistics	
	B	Std. Error	Beta			Tolerance	VIF
(Constant)	21.343	11.088		1.925	0.061		
Mean daily temperature (°C)	−0.058	0.451	−0.025	−0.128	0.899	0.364	2.749
Mean daily humidity (%)	0.158	0.151	0.187	1.052	0.299	0.438	2.281
Mean daily rainfall (inches)	0.251	0.314	0.132	0.799	0.429	0.506	1.975
Phase 2	−7.694	2.966	−0.415	−2.594	0.013	0.539	1.857
Phase 3	−14.453	4.780	−0.625	−3.024	0.004	0.323	3.100

Dependent variable: mean daily PM_2.5_ concentration, R = 65.9%, R^2^ = 43.5%, R^2^ (adjusted) = 36.6%.

## Data Availability

The datasets used and/or analyzed during the current study are available from the corresponding author on reasonable request.

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
