# Peer review of "The Impact of the Lockdown Caused by the COVID-19 Pandemic on the Fine Particulate Matter (PM2.5) Air Pollution: The Greek Paradigm"

_ijerph, 2021, doi:10.3390/ijerph18136748_

Round 1
Reviewer 1 Report
The authors were trying to study the impact of the COVID lockdown on the PM2.5 level in Greek, which is a hot topic and quite relevant to the scope of IJERPH. However, the manuscript still has some issues to fix before it can be publishable.
1), after reading the paper, I found there could be three issues with the title.
First, the COVID-19 pandemic cannot directly impact the air quality. In fact, the authors were trying to study the impact of the lockdown caused by the COVID-19 pandemic on the air quality.
Second, if the authors want to study the impact of COVID-19 lockdown on the air quality in Greek, the author must exclude other confounder factors, such as, meteorological factors and the contribution of air pollutants from neighboring countries that could experience lockdowns in a similar period. At the current stage, the authors only studied the change of air quality in the period of lockdown compared with the year 2019.
Third, “Particulate matter PM2.5 air pollution” is not appropriate, and “impact of sth. in sth.” could be grammatically wrong.
2). P1, Line 42. 2.5 btm?
3). P3, Line 123. Please specify the method. The sentence “The research team developed a mathematical formula and an optimization model” is very vague. What are the formula and model? Please specify.
4). Again, “A mathematical formula developed to divide the city into smaller areas..” Please specify the formula.
5). P5, Line 186. The author only compared the data during the lockdown to the previous year. However, lots of studies have used the data during the same period in the last five years. Especially, the air quality could be better and better across the years [Li and Federico, 2020] due to some reasons such as the government investment in the air quality improvement. I would suggest the authors to exam this possible confounder.
Li, J., & Tartarini, F. (2020). Changes in Air Quality during the COVID-19 Lockdown in Singapore and Associations with Human Mobility Trends. Aerosol and Air Quality Research, 20(8), 1748–1758.
6) The aim of Section 3.3 about the correlations between meteorological variables and PM2.5 is not clear. How can this part be relevant to the title of the paper? Maybe the authors can try to find correlations between PM2.5 and other parameters which could be potentially related to the lockdown, such as, the change of human mobility.
Author Response
RESPONSE TO COMMENTS FROM REVIEWER 1:
- The authors were trying to study the impact of the COVID lockdown on the PM5 level in Greek, which is a hot topic and quite relevant to the scope of IJERPH.
RESPONSE: We sincerely thank you for your kind words about our paper. We are delighted to receive positive feedback from you.
- However, the manuscript still has some issues to fix before it can be publishable.1), after reading the paper, I found there could be three issues with the title. First, the COVID-19 pandemic cannot directly impact the air quality. In fact, the authors were trying to study the impact of the lockdown caused by the COVID-19 pandemic on the air quality.
RESPONSE: Thank you for this remark. We agree and the title has been revised accordingly in the revised manuscript.
- Second, if the authors want to study the impact of COVID-19 lockdown on the air quality in Greek, the author must exclude other confounder factors, such as, meteorological factors and the contribution of air pollutants from neighboring countries that could experience lockdowns in a similar period. At the current stage, the authors only studied the change of air quality in the period of lockdown compared with the year 2019.
RESPONSE: Thank you for this great suggestion. We agree that the investigation of the contribution of air pollutants from neighboring countries that could experience lockdowns in a similar period would be interesting to know. The data that we have cannot reveal this, but our protocols could be modified for a follow-up study. Our study was designed to focus specifically on to what extent government restrictions affected PM2.5 concentrations in one of the most polluted cities in Greece, Volos, which constitutes an excellent paradigm of a medium-size city with significantly increased air pollution than neighboring cities. On the other hand, data regarding meteorological factors such as temperature, relative humidity, rainfall, and wind speed were also recorded along with each PM2.5 measurement by the network and were examined as confounders in our study.
- Third, “Particulate matter PM5 air pollution” is not appropriate, and “impact of sth. in sth.” could be grammatically wrong.
RESPONSE: Thank you for these points. We apologize for the errors. We have corrected them in the revised manuscript.
P1, Line 42. 2.5 btm?
RESPONSE: Thank you for this point. In the revision, we have defined this point.
- P3, Line 123. Please specify the method. The sentence “The research team developed a mathematical formula and an optimization model” is very vague. What are the formula and model? Please specify.
RESPONSE: Thank you for this comment. We specified the method on page 5, lines 198-225. More information regarding the methodology presented in this study is available on request. The data are not publicly available due to privacy.
- Again, “A mathematical formula developed to divide the city into smaller areas.” Please specify the formula.
RESPONSE: Thank you for this comment. We have introduced this data on page 5, lines 198-225.
- P5, Line 186. The author only compared the data during the lockdown to the previous year. However, lots of studies have used the data during the same period in the last five years. Especially, the air quality could be better and better across the years [Li and Federico, 2020] due to some reasons such as the government investment in the air quality improvemen I would suggest the authors to exam this possible confounder. [Li, J., & Tartarini, F. (2020). Changes in Air Quality during the COVID-19 Lockdown in Singapore and Associations with Human Mobility Trends. Aerosol and Air Quality Research, 20(8), 1748–1758.]
RESPONSE: You raise a very valid point. Air quality data provided by the Ministry of Environment, Energy And Climate Change in Greece are available only for the last four years (since 2017). The air quality has not been better across the years in Greece. The average values of PM2.5 since 2017 were similar to the average PM2.5 values of 2019 presented in the manuscript. No government investment in air quality improvement has been recorded (ref: https://ypen.gov.gr/perivallon/poiotita-tis-atmosfairas/dedomena-metriseon-atmosfairikis-rypansis/).
- The aim of Section 3.3 about the correlations between meteorological variables and PM2.5 is not clear. How can this part be relevant to the title of the paper?
RESPONSE: Thank you for this comment. Air quality is closely related to meteorological conditions, given that these variables affect pollution concentration, as well as the removal, transportation and dispersion of the airborne particles.
- Maybe the authors can try to find correlations between PM2.5 and other parameters which could be potentially related to the lockdown, such as, the change of human mobility.
RESPONSE: Thank you for this suggestion. Lockdown restrictions significantly impacted citizens’ mobility. 55 GPS devices installed in city buses demonstrate that, on average, traffic was decreased by 50% on the main roads of Volos. Unfortunately, no more detailed data are provided from local or national sources to make a more extensive analysis. However, this is a great suggestion for future research.
We appreciate you taking the time to offer us your insights related to the paper. We found your feedback very constructive. We tried to be responsive to your concerns.
Reviewer 2 Report
Review comments for the manuscript titled “The impact of Coronavirus disease 2019 pandemic in particulate matter PM2.5 air pollution: The Greek paradigm” submitted to IJERPH by Kotsiou et al.
This study examines ambient PM2.5 levels before, during and after the main COVID-19 shutdown at 12 locations in an industrial city in Greece. Meteorological parameters are also measured concurrently at each location. All the measurements are then compared to those corresponding levels in 2019. Finally the relation of PM2.5 levels to the changes in meteorological conditions and shutdown periods are assessed. Even the samples were collected at 12 different points, no comparison between the locations has been provided in this manuscript. Also it is not clear which data have been used to calculate the mean values and make the comparison between two years, i.e., 2019 vs. 2020. To my understanding, mean values of the measurements from all 12 locations are used through the manuscript. This makes all the discussion on the site selections etc. insignificant and unnecessary inside the manuscript. It can be prepared again by including pollution distribution maps or more discussion on site comparison based on the findings provided in Fig. 2. Unfortunately I do not recommend this manuscript to be published in IJERPH with this current version since it does not have any unique contribution to the literature other than providing local PM levels during the main COVID-19 shutdown. Authors can find my comments for each section below.
Title:
I suggest not using particulate matter and PM2.5 together here. “..fine particulate matter (PM2.5)” is a better expression.
Abstract:
Lines 19-20: Please specify the sampling period. It reads like the annual means here for 2020 and 2019. How did the authors make this comparison? I may miss the point but could not see any information inside the manuscript regarding where and how the 2019 data were collected. Also were the means calculated by averaging daily PM levels from all the 12 locations over the sampling duration? Please explain it clearly.
Line 21: ….exceeded the WHO’s guideline value for 24-hour mean PM2.5….
Lines 19-23: There is no explanation of three phases before mentioning phase 2 in line 22. So please provide the dates instead. One PM2.5 standard is used. So please make it singular in lines 21 and 22. I suggest providing dates for before, during and after the lockdown periods separately in line 19 first.
Keywords:
I suggest replacing “particulate matter” with PM2.5 or fine particulate matter.
Introduction:
Lines 35-37: Long sentence connected with several “and”s.
Line 40: … reactive oxygen species..
Line 43: What does “btm” mean here? Better definition of PM2.5 should be provided here.
Lines 38-60: The information discussed here is too general. Can be shortened since no elemental analysis or source apportionment analysis have been aimed in this study.
Line 57: Please correct the use PM2.5s here.
Lines 61-73: If necessary, this discussion can be moved to the materials & method section, somewhere describing the sampling area, sampling period, and meteorological measurements.
Line 89:…before, during, and after the …
Dates can be provided along with each period.
Materials and Methods:
Lines 98-100: Time intervals used for measuring meteorological parameters should be far smaller.
Line 118: replace “were” with “are”.
Line 119: What does “I/O” mean here? Indoor to outdoor ratio?
Lines 123-159: Large discussion of selecting the sampling location have been provided here but none of the result is site-specific. Is there any possibility to merge the information given in Fig.3 and Fig.4B? I would delete Fig. 4A since these are just the planned locations and no samples were collected in A-section. Were the ambient PM2.5 data collected from the identical locations during the same period in 2019? Please provide detailed information on 2019 sampling/sample location(s).
Lines 166-170: At which locations shown in Fig. 4B did the authors make this comparison? What does “reference instrument” mean here? Please provide some details about the reference instrument’s PM2.5 measurement method and mean and STDEV values for the comparisons.
Results:
Lines 183-184: Subtitle is too long. Better to say something like: Comparison between the ambient PM2.5 levels in 2020 and 2019
Line 188:… in Volos over the entire sampling period.
Figure 5: I suggest using markers instead of lines; then the readers can easily differentiate the daily samples. Also use tick marks in each axis. Vertical lines I assume that separate the phases. Instead of typing start dates of each phase above those areas, I suggest typing phase 1, phase 2.. there. If so, why was the separation for phase 1 started from March 10? Again please clearly specify the data source for both years here. Add “daily”.. to y-axis’s title.
Lines 200-201: Is there any explanation for this sharp drop at the end of April? Effect of domestic heating emissions or long-range transported dust emissions?
Lines 200-201 and 250-251: “In both years, reduction in the PM levels was..”
What does “successive periods” mean here?
Lines 202-2007: Daily PM levels in 2019 before the end of April are all above the WHO’s standard. So I do not think thank talking about a decrease in concentrations between P1 and P2 is appropriate (Some results before March 10 are even lower than those observed during these periods).
Lines 208-210: The number of samples < than the WHO’s standard value seems to be lower than 41% during shutdowns in 2020 when checking from the fig. 5.
Figure 6: I do not thinks that this figure makes a big contribution to the manuscript. A bar graph, comparing the PM levels over the same periods in 2019 and 2020 can be incorporated as a separate subfigure to Fig. 5.
Figure 7: Any explanation for the positive correlation between PM2.5 levels and rainfall? It is not typical
Discussion:
Lines 255, 291: “…contributed lower to PM2.5…”
Lines 257-285: should be discussed earlier in the manuscript. Can be moved to Introduction. How can sea salt make up 80%of PM2.5 levels? Please add a reference here.
I suggest authors to perform daily air mass trajectory analysis at one location at the study area over the sampling period in 2020 to be able to see whether the dust transport effects the PM levels or not.
Lines 292-294: Conflict with the assessment at the end of the previous paragraph. Lines 299-300 again conflicts with the observation in line 292-294, if I do not get confused.
Lines 307-309: Discussion on nanoparticles and health effect of these particulates is outside this manuscript’s content.
Lines 309-319: Move to Introduction if necessary.
Conclusions:
Lines 334-335: There is no strong proof from this study the authors can provide for this observation.
Lines 335-337: Again no toxicological assessment has been made for this discussion.
Author Response
RESPONSE TO COMMENTS FROM REVIEWER 2
- Review comments for the manuscript titled “The impact of Coronavirus disease 2019 pandemic in particulate matter PM2.5 air pollution: The Greek paradigm” submitted to IJERPH by Kotsiou et al. This study examines ambient PM2.5 levels before, during and after the main COVID-19 shutdown at 12 locations in an industrial city in Greece. Meteorological parameters are also measured concurrently at each location. All the measurements are then compared to those corresponding levels in 2019. Finally the relation of PM2.5 levels to the changes in meteorological conditions and shutdown periods are assessed.
RESPONSE: We appreciate all of your insightful comments. We found them quite useful as we approached our revision. We are grateful for the time and energy you expended on our behalf. In the following pages are our point-by-point responses to each of your comments.
- Even the samples were collected at 12 different points, no comparison between the locations has been provided in this manuscript.
RESPONSE: We greatly appreciate this comment. We agree that comparisons of air quality between 12 different city points would be interesting to know, but these data are under analysis, and so we will leave this information to another research.
- Also it is not clear which data have been used to calculate the mean values and make the comparison between two years, i.e., 2019 vs. 2020. To my understanding, mean values of the measurements from all 12 locations are used through the manuscript. This makes all the discussion on the site selections etc. insignificant and unnecessary inside the manuscript.
RESPONSE: Thank you for this valuable comment. The values of the 12 monitoring devices have been used to calculate the daily mean PM2.5 concentration in Volos. We consider that it is important to describe the selection areas in order to confirm that our measurements were representative of the whole city, given that the area of the municipality of Volos is 387.1 km2.
- It can be prepared again by including pollution distribution maps or more discussion on site comparison based on the findings provided in Fig. 2.
RESPONSE: We agree that the comparisons of air quality between 12 different city points would be an important line of study. We have now acknowledged this and suggested it as a topic for further research in the section of study limitations (page 11, lines 443-445).
- Unfortunately I do not recommend this manuscript to be published in IJERPH with this current version since it does not have any unique contribution to the literature other than providing local PM levels during the main COVID-19 shutdown. Authors can find my comments for each section below.
RESPONSE: Thank you for this comment. The contribution of this study was to provide evidence to what extent government restrictions affected PM2.5 concentrations in one of the most polluted cities in Greece, Volos. This is the first study conducted in Greece regarding this topic. We found your feedback very constructive. We tried to be responsive to your concerns.
- Title: I suggest not using particulate matter and PM2.5 together here. “..fine particulate matter (PM2.5)” is a better expression.
RESPONSE: Thank you for this point. We have revised the title accordingly.
- Abstract: Lines 19-20: Please specify the sampling period. It reads like the annual means here for 2020 and 2019.
RESPONSE: Thank you for this comment. In the revision, we have specified the sampling period as suggested (page 1, lines 16-17).
- How did the authors make this comparison? I may miss the point but could not see any information inside the manuscript regarding where and how the 2019 data were collected. Also were the means calculated by averaging daily PM levels from all the 12 locations over the sampling duration? Please explain it clearly.
RESPONSE: Thank you for this remark. We apologize for the misunderstanding. We clarified this issue on page 4, lines 179-183.
- Line 21: ….exceeded the WHO’s guideline value for 24-hour mean PM2.5….
RESPONSE: Thank you for this point. We revised the sentence accordingly.
- Lines 19-23: There is no explanation of three phases before mentioning phase 2 in line 22. So please provide the dates instead. One PM2.5 standard is used. So please make it singular in lines 21 and 22. I suggest providing dates for before, during and after the lockdown periods separately in line 19 first.
RESPONSE: Thank you for the valuable comments and suggestions. We have revised the abstract accordingly (page 1, lines 14-28).
- Keywords:I suggest replacing “particulate matter” with PM2.5 or fine particulate matter.
RESPONSE: Thank you for this point. We have replaced the keyword as suggested.
- Introduction: Lines 35-37: Long sentence connected with several “and”s.
RESPONSE: Thank you for this point. We have revised the sentence, as suggested.
- Line 40: … reactive oxygen species..
RESPONSE: We apologize for the omission. We have corrected the phrase (page 2, line 57).
- Line 43: What does “btm” mean here? Better definition of PM2.5 should be provided here.
RESPONSE: Thank you for this point. In the revision, we have defined this point.
- Lines 38-60: The information discussed here is too general. Can be shortened since no elemental analysis or source apportionment analysis have been aimed in this study.
RESPONSE: Thank you for this remark. Per you suggestion, we have revised this section accordingly.
- Line 57: Please correct the use PM2.5s here.
RESPONSE: Thank you for this point. We have corrected this phrase.
- Lines 61-73: If necessary, this discussion can be moved to the materials & method section, somewhere describing the sampling area, sampling period, and meteorological measurements.
RESPONSE: Thank you for this point. This discussion has been moved to the materials and methods.
- Line 89:…before, during, and after the …
RESPONSE: Thank you for this point. We have corrected this phrase.
- Dates can be provided along with each period.
RESPONSE: Thank you for this point, we have provided dates per your suggestion.
- Materials and Methods: Lines 98-100: Time intervals used for measuring meteorological parameters should be far smaller.
RESPONSE: Thank you for this comment. The wireless sensors network was designed to measure the 24-h temperature, relative humidity, rainfall, and wind speed values every 2 seconds along with each PM2.5 measurement. We clarified this issue on page 4, line 167.
- Line 118: replace “were” with “are”.
RESPONSE: Thank you for this point. We have replaced “were” with “are”.
- Line 119: What does “I/O” mean here? Indoor to outdoor ratio?
RESPONSE: Thank you for this point. I/O refers to Zero-one integer. In the revised manuscript, we explained this abbreviation.
- Lines 123-159: Large discussion of selecting the sampling location have been provided here but none of the result is site-specific.
RESPONSE: Thank you for this comment. As we have previously mentioned, we consider that it is important to describe the selection areas in order to confirm that our measurements were representative of the whole city.
- Is there any possibility to merge the information given in Fig.3 and Fig.4B? I would delete Fig. 4A since these are just the planned locations and no samples were collected in A-section.
RESPONSE: Thank you for this comment. The legend of Figure 4 has been revised properly.
- Were the ambient PM2.5 data collected from the identical locations during the same period in 2019? Please provide detailed information on 2019 sampling/sample location(s).
RESPONSE: The ambient PM2.5 data were collected from the identical locations during the same period in 2019 as discussed in page 4, lines 179-18.
- Lines 166-170: At which locations shown in Fig. 4B did the authors make this comparison? What does “reference instrument” mean here? Please provide some details about the reference instrument’s PM2.5 measurement method and mean and STDEV values for the comparisons.
RESPONSE: Initially, the sensing measurements were performed in the lab by using the reference instrument. The second phase of sensors testing took place at each location by the same reference instrument. The reference instrument used followed the standards of the Ministry of the Environment and the European Commission for air quality monitoring.
- Results: Lines 183-184: Subtitle is too long. Better to say something like: Comparison between the ambient PM2.5 levels in 2020 and 2019
RESPONSE: The subtitle has been revised accordingly.
- Line 188:… in Volos over the entire sampling period.
RESPONSE: The sentence has been revised accordingly.
- Figure 5: I suggest using markers instead of lines; then the readers can easily differentiate the daily samples. Also use tick marks in each axis. Vertical lines I assume that separate the phases. Instead of typing start dates of each phase above those areas, I suggest typing phase 1, phase 2.. there. If so, why was the separation for phase 1 started from March 10? Again please clearly specify the data source for both years here. Add “daily”.. to y-axis’s title.
RESPONSE: Thank you for this comment. In the revision we have introduced a new Figure 5, per your suggestions.
- Lines 200-201: Is there any explanation for this sharp drop at the end of April? Effect of domestic heating emissions or long-range transported dust emissions?
RESPONSE: We are thankful for this comment. We discuss this issue on page 11, lines 400-407.
- Lines 200-201 and 250-251: “In both years, reduction in the PM levels was..”
RESPONSE: Thank you for this point. We have revised the sentences.
- What does “successive periods” mean here?
RESPONSE: We apologize for the misunderstanding. We mean the consecutive phases of the study period.
- Lines 202-2007: Daily PM levels in 2019 before the end of April are all above the WHO’s standard. So I do not think thank talking about a decrease in concentrations between P1 and P2 is appropriate (Some results before March 10 are even lower than those observed during these periods).
RESPONSE: Thank you for this suggestion. We agree with you as a borderline statistical significance was detected in PM2.5 levels between phase 1 (correspond to the COVID-19 pre-lockdown period) and phase 2 (correspond to the COVID-19 post-lockdown period) (43.65±7.72 vs. 36.6±9.72, p=0.046). Hence, we have removed this sentence, as suggested.
- Lines 208-210: The number of samples < than the WHO’s standard value seems to be lower than 41% during shutdowns in 2020 when checking from the fig. 5.
RESPONSE: Thank you for this comment. During the strictest lockdown (23th March to 4th May), the mean daily PM2.5 levels exceeded the standard 41% of the time.
- Figure 6: I do not thinks that this figure makes a big contribution to the manuscript. A bar graph, comparing the PM levels over the same periods in 2019 and 2020 can be incorporated as a separate subfigure to Fig. 5.
RESPONSE: Thank you for this point. Figure 6 has been removed.
- Figure 7: Any explanation for the positive correlation between PM2.5 levels and rainfall? It is not typical
RESPONSE: Thank you for this interesting point. We discussed this issue on page 11, lines 416-423.
- Discussion: Lines 255, 291: “…contributed lower to PM2.5…”
RESPONSE: We apologize for these errors. In the revision, we have corrected them accordingly.
- Lines 257-285: should be discussed earlier in the manuscript. Can be moved to Introduction. How can sea salt make up 80% of PM2.5 levels? Please add a reference here.
RESPONSE: Thank you for this comment. This part has been moved to Introduction, as recommended. Sea salt is one of the major atmospheric aerosol components (Lewis and Schwartz, 2004) and can significantly impact particulate matter concentrations. Sea-salt aerosol (SSA) is mainly produced by bursting bubbles during whitecap formation in the open-ocean (Monahan et al.,1986). A more localized mechanism of SSA production involves waves breaking in the surf zone. The diameter of SSA ranges from less than 0.2µm to greater than 2000µm. The ambient SSA mass distribution is dominated by particles in the 1–10µm diameter range. Particles smaller than 1µm can make substantial contributions to cloud condensation nuclei concentrations (Pierce and Adams, 2006), while particles larger than 50µm have a very short atmospheric lifetime and play a negligible role in atmospheric chemistry (Lewis and Schwartz, 2004). SSA participates in atmospheric chemistry by interacting with atmospheric pollutants.
- I suggest authors to perform daily air mass trajectory analysis at one location at the study area over the sampling period in 2020 to be able to see whether the dust transport effects the PM levels or not.
RESPONSE: Thank you for this great suggestion, which could be a subject for a following study. Unfortunately, the deadline for revisions put a bit of pressure on our team to perform such an analysis in a reasonable time.
- Lines 292-294: Conflict with the assessment at the end of the previous paragraph. Lines 299-300 again conflicts with the observation in line 292-294, if I do not get confused.
RESPONSE: Thank you for the comments and we apologize for the misunderstanding. The sentences have been revised to be more comprehensible to readers.
- Lines 307-309: Discussion on nanoparticles and health effect of these particulates is outside this manuscript’s content. Lines 309-319: Move to Introduction if necessary.
RESPONSE: Thank you for this comment. This part has been moved to introduction, as suggested.
- Conclusions: Lines 334-335: There is no strong proof from this study the authors can provide for this observation.
RESPONSE: Thank you for this remark. We have removed this sentence as reccomended.
- Lines 335-337: Again no toxicological assessment has been made for this discussion.
RESPONSE: Thank you for this point. We have removed this sentence as suggested.
We are very grateful for the effort you dedicated to reviewing our submission, as well as your favorable comments. We paid heed to your advice and suggestions, and the manuscript has been massively revised. We hope the revised version of our manuscript will satisfy your concerns.
Round 2
Reviewer 1 Report
The authors have made item-by-item responses to my comments and revised the manuscript in detail. The manuscript has been improved a lot, only left one point for you to further consider.
Point 9 based on the last comments about the human mobility data. The author replied that it is hard to find the related data available. However, both Google and Apple have open databases about that. Please find the detailed information in the paper below. Please consider to at least discuss it about that.
[Li, J., & Tartarini, F. (2020). Changes in Air Quality during the COVID-19 Lockdown in Singapore and Associations with Human Mobility Trends. Aerosol and Air Quality Research, 20(8), 1748–1758.]
Author Response
Reviewer 1.
- The authors have made item-by-item responses to my comments and revised the manuscript in detail. The manuscript has been improved a lot, only left one point for you to further consider.
RESPONSE: We sincerely thank you for your kind words about our paper. We are delighted to receive positive feedback from you.
- Point 9 based on the last comments about the human mobility data. The author replied that it is hard to find the related data available. However, both Google and Apple have open databases about that. Please find the detailed information in the paper below. Please consider to at least discuss it about that. [Li, J., & Tartarini, F. (2020). Changes in Air Quality during the COVID-19 Lockdown in Singapore and Associations with Human Mobility Trends. Aerosol and Air Quality Research, 20(8), 1748–1758.
RESPONSE: Thank you for this direction. We have searched the databases from Google and Apple about traffic congestion levels provided data only for the two Greece's largest cities, Athens and Thessaloniki. In Athens and Thessaloniki, there was 21% and 27% less traffic in 2020 than in 2019, respectively, as shown in the following figure (ref. https://www.tomtom.com/en_gb/traffic-index/greece-country-traffic/). Per your suggestion, we have introduced this data in the main manuscript (page 9, lines 320-324).
We are very grateful for the effort you dedicated to reviewing our submission, as well as your favorable comments. We hope the revised version of our manuscript will satisfy your concerns.

Reviewer 2 Report
Some small points that can be corrected:
Line 60: ...referred to as inhalable fine particulate matter (PM2.5).
Lines 65-666: Please delete the sentence "PM2.5 pollution can be either human-made or naturally occurring."
Lines 68-69: "Human-made sources of PM2.5 are more important than natural sources which make only a small contribution to the total concentration." is a weak statement since there are many variables affecting PM composition at a particular location.
Lines 75-79: Please add references if there are any publications on ambient PM sources in the study area.
Lines 80-118: Big discussion on sources of PM in the study area. To me they can be discussed within max one or two paragraphs only with some references since this is not a source apportionment study and the results do only provide the mass difference.
Lines 119-121: Can be deleted since no nanoparticle issue has been studies.
Line 149: ...located at...
Line 153: ...those obtained from the identical locations during the same period in 2019...
Fig. 5: What does vertical dash line near March 23 show?
Line 270: At the beginning of the result section I would add something like "Daily PM2.5 concentrations used in Fig. 5 and throughout this study were obtained by averaging 24-hour PM2.5 measurements monitored at12 locations in 2019 and 2020 ."
Figure 6, c: It is not typical to get positive correlations between the rainfall and PM levels. There should be negative correlation instead. So please make a brief explanation. I would remove this figure.
Author Response
Reviewer 2.
- Some small points that can be corrected: Line 60: ...referred to as inhalable fine particulate matter (PM2.5).
RESPONSE: Thank you for this point. We have revised the sentence, as suggested.
- Lines 65-66: Please delete the sentence "PM2.5 pollution can be either human-made or naturally occurring."
RESPONSE: Thank you for this comment. Per your suggestion, we have deleted the sentence.
- Lines 68-69: "Human-made sources of PM2.5 are more important than natural sources which make only a small contribution to the total concentration." is a weak statement since there are many variables affecting PM composition at a particular location.
RESPONSE: Thank you for this comment. Per your suggestion, we have removed this sentence.
- Lines 75-79: Please add references if there are any publications on ambient PM sources in the study area.
RESPONSE: Thank you for this remark. We have added the proper references.
- Lines 80-118: Big discussion on sources of PM in the study area. To me they can be discussed within max one or two paragraphs only with some references since this is not a source apportionment study and the results do only provide the mass difference.
RESPONSE: Thank you for this comment. We have limited the discussion, as suggested.
Lines 119-121: Can be deleted since no nanoparticle issue has been studies.
RESPONSE: Thank you for this point. We have deleted this sentence, as recommended.
- Line 149: ...located at...
RESPONSE: We apologize for the error. In the revision, we have corrected it.
- Line 153: ...those obtained from the identical locations during the same period in 2019...
RESPONSE: Thank you for this comment. We have revised the sentence as suggested.
- 5: What does vertical dash line near March 23 show?
RESPONSE: Thank you for this comment. The vertical black dashed line represents the initiation of the strictest period amid phase 2 (March 23rd, 2020).
- Line 270: At the beginning of the result section I would add something like "Daily PM2.5 concentrations used in Fig. 5 and throughout this study were obtained by averaging 24-hour PM2.5 measurements monitored at12 locations in 2019 and 2020 ."
RESPONSE: Thank you for this suggestion. We have added this sentence as proposed.
- Figure 6, c: It is not typical to get positive correlations between the rainfall and PM levels. There should be negative correlation instead. So please make a brief explanation. I would remove this figure.
RESPONSE: Thank you for this suggestion. Indeed, the correlation between PM2.5 levels and rainfall is still unclear. We have removed Figure 1C, as suggested.